# High-Performance Self-Powered Ultraviolet Photodetector Based on Nano-Porous GaN and CoPc *p–n* Vertical Heterojunction

**DOI:** 10.3390/nano9091198

**Published:** 2019-08-26

**Authors:** Yan Xiao, Lin Liu, Zhi-Hao Ma, Bo Meng, Su-Jie Qin, Ge-Bo Pan

**Affiliations:** 1Key Laboratory of Optoelectronic Devices and Systems of Ministry of Education and Guangdong Province, College of Optoelectronic Engineering, Shenzhen University, Shenzhen 518060, China; 2Suzhou Institute of Nano-tech and Nano-bionics, Chinese Academy of Sciences, Suzhou 215123, China; 3Department of Health and Environmental Sciences, Xi’an Jiaotong-Liverpool University, Suzhou 215123, China

**Keywords:** ultraviolet photodetectors, nano porous-GaN, organic/inorganic hybrids, *p–n* heterojunction, self-powered

## Abstract

Gallium nitride (GaN) is a superior candidate material for fabricating ultraviolet (UV) photodetectors (PDs) by taking advantage of its attractive wide bandgap (3.4 eV) and stable chemical and physical properties. However, the performance of available GaN-based UV PDs (e.g., in terms of detectivity and sensitivity) still require improvement. Fabricating nanoporous GaN (porous-GaN) structures and constructing organic/inorganic hybrids are two effective ways to improve the performance of PDs. In this study, a novel self-powered UV PD was developed by using *p*-type cobalt phthalocyanine (CoPc) and *n*-type porous-GaN (CoPc/porous-GaN) to construct a *p–n* vertical heterojunction via a thermal vapor deposition method. Under 365 nm 0.009 mWcm^−2^ light illumination, our device showed a photoresponsivity of 588 mA/W, a detectivity of 4.8 × 10^12^ Jones, and a linear dynamic range of 79.5 dB, which are better than CoPc- and flat-GaN (CoPc/flat-GaN)-based PDs. The high performance was mainly attributed to the built-in electric field (BEF) generated at the interface of the CoPc film and the nanoporous-GaN, as well as the nanoporous structure of GaN, which allows for a higher absorptivity of light. Furthermore, the device showed excellent stability, as its photoelectrical property and on/off switching behavior remained the same, even after 3 months.

## 1. Introduction

Ultraviolet (UV) detection is essential in various technical fields including applications in flame sensing, environmental monitoring, and spatial optical communications [1,2,3,4]. To date, various UV photodetectors (PDs) using different structures, such as Schottky junctions [5,6], metal–semiconductor–metal sandwich structures [7,8], *p–n* junctions [9,10,11], and photoelectrochemical cells [12,13,14], have been proposed. Among these structures, *p–n* heterojunctions are the most effective because of the improved separation efficiency of photoexcited electron–hole pairs by the built-in electric field (BEF) [15,16,17]. Furthermore, the photovoltaic effect of *p–n* heterojunctions can enable the PDs to become self-powered devices, and a great deal of work has been done on making self-powered UV PDs on the basis of *p–n* junctions [18,19]. Si-based photodiodes are one of most popular commercially available UV PDs. However, due to the narrow band gap (~1.12 eV) and instability of Si, additional filters are needed in the Si-based photodiodes, which increases the costs and volume of PDs [20,21,22,23,24]. 

Gallium nitride (GaN) has excellent properties, such as direct wide bandgap (3.4 eV), high electron mobility, large saturation velocity, and good thermal stability, and has been widely used for fabricating UV PDs [25,26,27]. Although many impressive GaN-based devices have been constructed [28,29,30], the performance of available GaN-based UV PDs is still unsuitable for real applications due to their low photo detectivity and quantum efficiency, resulting from low light absorption and the quick recombination of photo-generated electron–hole pairs [31]. Fabricating nanoporous GaN (porous-GaN) is a promising approach to improve the light absorption by providing large specific surface area from nanopores [9,32,33,34]. These unique porous structures can also serve as photo traps to increase light absorptivity. 

To date, most inorganic UV PDs are based on single photoconductive materials, such as III–V group semiconductors, metal oxide semiconductors, and SiC, because stable *p*-type inorganic materials and inorganic *p–n* heterojunction devices are difficult to fabricate [35]. Thus, constructing organic/inorganic heterojunctions is a promising way to fabricate self-powered PDs, because *p*-type organic semiconductors are very popular due to their low cost and easy fabrication process [36]. Organic/inorganic hybrid structures can combine the unique properties of organic and inorganic materials in making UV PDs [37]. In 2017, Zhou et al. reported a high-performance UV PD based on CH_3_NH_3_PbI_3_/GaN hybrids; the detectivity reached 7.96 × 10^12^ Jones at 0 V [38]. However, considering the instability of CH_3_NH_3_PbI_3_ in air, more stable organic materials are needed in UV PD preparation. Among the reported organic materials, organic small conjugated molecules which have tunable optical property and high mobility have been extensively studied in recent years [39]. For instance, metal phthalocyanines (MPcs) are some of the most widely studied organic functional materials due to their unique electrical and optical properties as well as their good chemical and thermal stability [40,41,42]. CoPc is a typical *p*-type organic semiconductor material of the MPc family, which has an 18 π-electron system with Co^2+^ in the center (Appendix A) [43]. Considering its wide band gap (3.35 eV) and high carrier mobility [44], *p*-type CoPc is a suitable alternative organic material to construct *p–n* heterojunction UV PDs with *n*-type GaN.

In this study, we designed a novel high-performance self-powered UV PD based on an organic/inorganic hybrid *p–n* heterojunction. Specific organic small conjugated CoPc molecules which have wide band gap (3.35 eV) and high thermal stability were used as the *p*-type material, and GaN with nanopores was employed as the *n*-type material. A *p–n* heterojunction was formed at the interface of porous-GaN and CoPc to enhance the sensitivity and form a self-powered PD. The large specific surface area of the unique nanoporous structure not only improves light absorption, but also serves as photo traps to increase light absorption. The proposed device exhibited higher responsivity (R), more specific detectivity (D*), larger switch ratio (I_on_/I_off_), and wider linear dynamic range (LDR) at 0 V bias to UV light compared to other single GaN-based or inorganic/inorganic hybrid based UV PDs. Furthermore, due to the high thermal stability of GaN and CoPc, the photoelectrical property and on/off switching behavior of our device remained the same after 3 months. This high performance and self-powered capability PD has potential application in many areas, including UV radiation surveillance, water quality monitoring, and spatial optical communications.

## 2. Materials and Methods 

### 2.1. Materials and Chemicals

The commercially available *n*-type flat GaN was obtained by doping Si in GaN and growing on sapphire (0001) substrates via hydride vapor phase epitaxy. Other materials were as follows: CoPc (99.99%, Aldrich Co., Ltd, Shanghai, China.), 1-butyl-3-methylimidazolium perchlorate ((BMIM)ClO_4_, 99%, Shanghai Cheng Jie Chemical Co., Ltd. Shanghai, China), sulfuric acid (H_2_SO_4_, 98% Sinopharm Chemical Reagent Co., Ltd, Suzhou, China), acetone (C_3_H_6_O, 99%, Sinopharm Chemical Reagent Co., Ltd, Suzhou, China), and ethanol (C_2_H_5_OH, 99%, Sinopharm Chemical Reagent Co., Ltd, Suzhou, China ). The flat GaN was cleaned by sonication in acetone, ethanol, and deionized water (>18 MX), respectively.

### 2.2. Preparation of Nanoporous GaN

The porous-GaN film was prepared by our previous reported work, through a photo-assisted electrochemical etching method [45]. Briefly, porous-GaN was prepared in a three-electrode cell system. The working electrode, working counter electrode, reference electrode, and etchant were flat GaN substrate, Pt wires, Pt wires, and (BMIM)ClO_4_ respectively. After applying a constant voltage of 3 V for 20 min, the as-etched flat GaN was immersed in aqua regia and deionized water, respectively.

### 2.3. Device Fabrication

Figure 1 presents the fabrication process schematic of the porous-GaN-based *p–n* heterojunction UV PD. Nanoporous GaN was obtained by photo-assisted electrochemical etching and used as substrate to grow the CoPc film. Then, the CoPc thin film (20 nm) was deposited by conventional thermal evaporation [46]. Briefly, CoPc powder was filled in a quartz crucible at a vacuum of 10^−4^ Pa. Thin CoPc films were obtained by heating the source with a constant deposition rate of 7 Ås^−1^. The substrate temperature was maintained at room temperature. The CoPc/porous-GaN heterojunction device was completed by thermal evaporation of 80-nm Au electrodes on the surface of the CoPc thin film and nanoporous GaN film, respectively.

### 2.4. Characterization

The as-prepared film was investigated by scanning electron microscopy (SEM, Hitachi S4800, Tokyo, Japan), Energy Dispersive X-ray Fluorescence (EDX) analyses were performed using a Hatchi s-4800 field emission scanning electron microscope (Hitachi, Tokyo, Japan) UV–Vis absorption spectra lambda750 (PerkinElmer Inc, Shanghai, China), and Raman spectra HR 800 (HORIBA JobinYvon, JobinYvon, France). The electrical characteristics of device were investigated by Keithley 4200 SCS (Tektronix Inc., Beaverton, OR, USA). The device was placed on a probe station in a clean and shielded box. The 365-nm UV light source with adjustable power density was provided by UVEC-4 (Shen Zhen Lamplic Tech Co., Shenzhen, China). All measurements were performed in ambient conditions at room temperature.

## 3. Results

### 3.1. Microstructures of the Nanoporous GaN and CoPc/Porous-GaN Film

Figure 2a displays a typical SEM image of as-prepared porous-GaN. Unlike flat GaN, which has a flat and smooth surface (Appendix A), 3D porous structures were formed on the entire surface. Each hole had a hexagonal cross-section with different size. One hundred pores were randomly selected for statistical analysis of size. The results given in Figure 2b show that among these 100 pores, there were no pores smaller than 20 nm or larger than 110 nm. The morphology of the CoPc film after deposition on porous-GaN is shown in Figure 2c. Compared with pure nanoporous GaN, numerous CoPc nanoparticles were deposited on the porous-GaN surface and cross-linked together to form a dense CoPc film. The black spots on the surface were mainly caused by the roughness of hole structure that led to electron beam shielding after CoPc deposition. The CoPc film was directly stacked on the porous-GaN (inset in Figure 2c), with partial penetration into the porous-GaN. Figure 2d shows the EDX spectra of pure porous-GaN and CoPc/porous-GaN film, the peaks of two elements (Ga and N) were observed in the spectra of pure porous-GaN. Meanwhile, the peaks of C, N, and Co, which came from CoPc, were observed in the spectra of CoPc/porous-GaN film. This indicated that CoPc was successfully deposited on the porous-GaN surface.

### 3.2. Optical Properties of the Nanoporous GaN and CoPc/Porous-GaN Film

The Raman spectra (Figure 3a) of both CoPc/porous-GaN and CoPc powder were recorded to confirm the chemical structure of CoPc film after being deposited on porous-GaN. Characteristic peaks of the CoPc film were identical to the CoPc powder, indicating no chemical modification of the CoPc film. The optical properties of the CoPc/porous-GaN film were also studied by UV–Vis absorbance. The absorption curves (shown in Figure 3b) show that both CoPc/flat-GaN and CoPc/porous-GaN films exhibited high absorptivity in the 300–400 nm range (especially at 320–380 nm). More importantly, the absorptivity of the CoPc/porous-GaN film was higher than that of the CoPc/flat-GaN film. The higher absorption of the CoPc/porous-GaN film was caused by the nanoporous structure of GaN, because a nanoporous structure can reduce the reflectivity of GaN, as reported in [9].

### 3.3. Device Characterization

The current–voltage (I–V) curves of PD in the dark and with different power densities of 365 nm light illumination are shown in Figure 4a. Typical rectification characteristics were exhibited in our device, suggesting that a well-behaved diode structure was formed due to the formation of *p–n* vertical heterojunction between *p*-type CoPc film and *n*-type porous-GaN. The current gradually increased with the increase of power density at the same voltage, suggesting that our device could be used in UV light detection. Notably, the current in the dark state was only 6.8 nA, whereas the photocurrent reached 71.8 µA under 1.4 mW/cm^2^ light illumination at 0 V bias. Figure 4b shows the photovoltaic behavior of our device. Our device exhibited an open circuit voltage (VOC) of 1.4 V, a short-circuit current (I_SC_) of 71.8 µA, and a fill factor (EF) of 0.26. The power conversion efficiency (η) of our device was ~3.6%, which is better than some relevant organic/inorganic hybrid devices [35,47]. Hence, this CoPc/porous-GaN *p*–*n* vertical heterojunction device could be applied for UV light detection with self-power ability.

The self-powered photo-response properties were investigated by adjusting UV light at 0 V bias. Figure 5a shows the photocurrent transient measurement by periodically switching 365 nm light using different power densities. The dark current reached the detectable limit of our instrument, and the photocurrent rapidly increased from the “OFF” level to the “ON” level after light illumination. The maximum value of the switch ratio was more than 10^5^ under 1.4 mW/cm^2^ light illumination. The photocurrent values remained constant (relative standard deviation <0.4%) after eight cycles, suggesting the high repeatability of the CoPc/porous-GaN *p-n* vertical heterojunction device. Device stability is an important factor for real applications. The proposed PD was exposed in air without any encapsulation and tested after several months to establish its stability. The time-dependent response curves of the device with 365 nm 0.009 mWcm^−2^ illumination at day one and after 3 months are shown in Appendix A. The proposed device demonstrated good photoelectrical property and remarkable on/off switching behavior. The photocurrent decreased only by 3.4% after 3 months. Compared with previously reported PDs [22,48], our device showed excellent stability due to the excellent thermal and chemical stability both of porous-GaN and CoPc.

Response time is also an important parameter in the performance quantification of PDs. The response time was obtained by enlarging the view of one dynamic response toward 1.4 mWcm^−2^ from Figure 5a. The rising and recovery times were defined as the time needed to go from 10% of the dark current to 90% of the photocurrent, or vice versa. The rising and recovery time were determined as 0.71 s and 0.5 s, respectively (Figure 5b). Meanwhile, the photocurrent increased with increase of the power density, further suggesting that current was highly dependent on power density. Figure 5c shows a linear relationship between the photocurrent and power density. This dependence relationship was fitted with power law as *I_p_* ~ *P^θ^*. From the curving fitting data, *θ* (which determines the response of the photocurrent to the power density) was 0.93, showing the existence of a carrier trap between Fermi level and conduction band edge [49].

The sensitivity to light is very important to PDs in practical applications. R and D* are two key parameters used to quantify the sensitivity, and can be expressed as: R = I_p_/(PA),(1)
D* = A^1/2^R/(2eI_d_)^1/2^,(2)
where R, I_p_, P, A, e, and I_d_ are the responsivity, photocurrent, power density, device area, elementary charge (1.60 × 10^−19^ C), and dark current of the PD, respectively. Figure 5d shows the power density dependence of R and D*. From this Figure, both R and D* rose sharply with decreasing power density when its value was lower than 0.1 mWcm^−2^. The maximum R was found to be 588 mA W^−1^ at 0.009 mWcm^−2^. This is larger than the most reported self-powered GaN-based UV PDs (Table 1). D* is another important parameter in quantifying the detection capability of PDs. The value of D* in this work (4.8 × 10^12^ Jones) also exhibits better performance than similar GaN-based and organic/inorganic hybrid PDs. The LDR is used to evaluate the linearity of the photosensitivity at various power densities, and can be expressed as [11]:LDR = 20 log(I_p_*/I_d_),(3)
where Ip* is the photocurrent of 1 mWcm^−2^ light illumination. From the equation, the LDR was 79.5 dB at 0 V, which was higher than 42 dB of a WS_2_/*n*-Si-based PD, 66 dB of a commercial InGaAs PD [50,51], and 77.51 dB of a GaSe/GaSb-based PD [52].

To compare with other reported works, some features of self-powered GaN-based UV PDs are summarized in Table 1. From this table, the responsivity of our device was the highest among the listed works. The detectivity was about five-fold lower than Zhou’s work, but demonstrated similar or better level than other works. For the switch ratio, our device could reach the best order of magnitude among those in the table. Therefore, our device exhibited good performance in terms of responsivity, detectivity, and switch ratio.

### 3.4. Sensing Mechanism

Energy-band diagrams which determine the photoresponse of our device are demonstrated in Figure 6. A *p–n* heterojunction was formed between the CoPc film and porous-GaN substrate due to their Fermi level difference. This difference leads the electrons in *n*-type porous GaN film to move to the *p*-type GoPc film, while the holes in the *p*-type CoPc film transferred to the *n*-type porous GaN film. This resulted in upward and downward bending of energy levels near the CoPc and porous GaN surface, respectively, until their energy levels reached the same level [55]. Finally, the BEF which was formed at the CoPc/porous-GaN interface would rapidly separate the photogenerated electron–hole pairs under UV light illumination, and induce the separated electrons and holes to transfer to the opposite electrodes. Therefore, the BEF at the heterojunction interface enabled the device to operate without any applied voltage.

### 3.5. Performance Enhancement Mechanism

For comparison, the performance of a device based on flat GaN was investigated. Appendix A shows the schematic structure of a PD based on a CoPc/flat-GaN *p–n* vertical heterojunction. The I–V curves of CoPc/flat-GaN-based PD in the dark and with different power densities of 365 nm light illumination are shown in Appendix A. The dark current was close to zero, meanwhile the photocurrent was clearly higher than the dark current and increased as the power density increased, suggesting that the CoPc/flat-GaN-based device could also be used in UV light detection. However, the response time was more than 5 s and continually increased as the device was cycled (Appendix A) for the CoPc/flat-GaN-based PD. In addition, the photocurrent of the CoPc/flat-GaN device was much lower than that of the CoPc/porous-GaN-based device, considering its dark current had approximately the same value, especially at zero (Figure 7), inducing a much smaller switch ratio of the CoPc/flat-GaN based PD. The above results indicate that the performance of the CoPc/flat-GaN-based PD was very inferior to the CoPc/porous-GaN-based PD, and the porous structure of the porous-GaN had a significant effect on the photoresponse enhancement. The improved performance was ascribed to the larger contact area between the CoPc film and the porous-GaN, and the higher absorptivity of porous-GaN. 

Compared with flat structures, porous-GaN has a larger specific surface area, which greatly enhances its sensitivity because the porous structure can serve as photo traps and decease reflectivity [56]. During the electrochemical etching process, pores preferentially initiate at the defect points of the flat GaN surface because of their higher chemical activity compared to other surface sites [42,57], leading to significant reduction of surface defects, which enhances the carrier mobility of the porous-GaN. As a result, the response speed of the proposed device was greatly increased. In addition, the response speed was further improved by the short interface migration distance, because CoPc nanoparticles are partially penetrated into the holes of porous-GaN and deposited on the surface of pore walls [31,58]. 

## 4. Conclusions

In this paper, we fabricated a novel UV PD with self-powered ability on the basis of CoPc/porous-GaN *p–n* heterojunction. The proposed PD exhibited high switch ratio (~10^5^), R (588 mA/W), and LDR (79.5 dB) that were much better than reported works, meanwhile, a D* of 4.8 × 10^12^ Jones displays an upper level among the reported work. It can be attributed to the nanopores on the GaN surface providing higher absorptivity of light and lower interface migration distance, which induced more electrons to take part in the excitation and recombination process. Given its excellent performance, zero energy consumption, and facile fabrication, our CoPc/porous-GaN *p–n* vertical heterojunction device can be employed in many applications, such as missile warning systems where high detectivity and stability of UV PDs are required.

## Figures and Tables

**Figure 1 nanomaterials-09-01198-f001:**
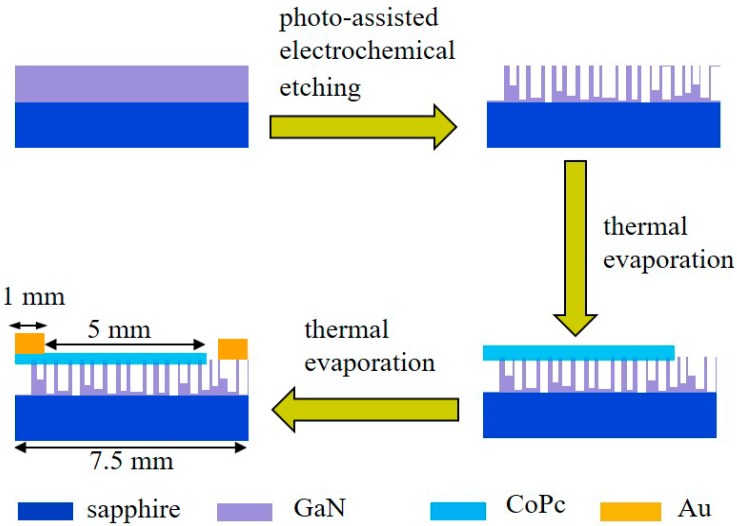
Schematic of the fabrication procedure of CoPc/ porous-GaN *p–n* vertical heterojunction. porous-GaN: nanoporous GaN.

**Figure 2 nanomaterials-09-01198-f002:**
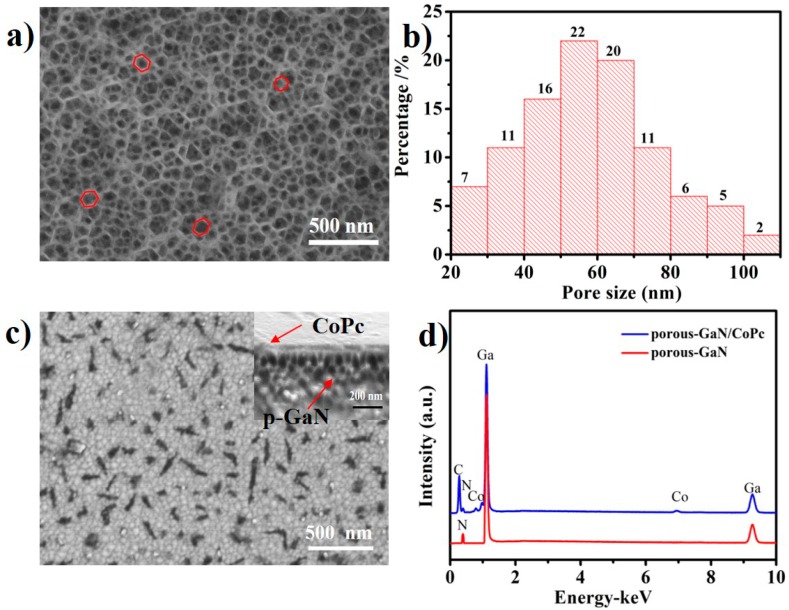
SEM images of (**a**) pure nanoporous GaN and (**b**) the corresponding pore size distribution diagram; (**c**) CoPc/porous-GaN film (inset is the cross section); (**d**) EDX spectra of porous-GaN and CoPc/porous-GaN film.

**Figure 3 nanomaterials-09-01198-f003:**
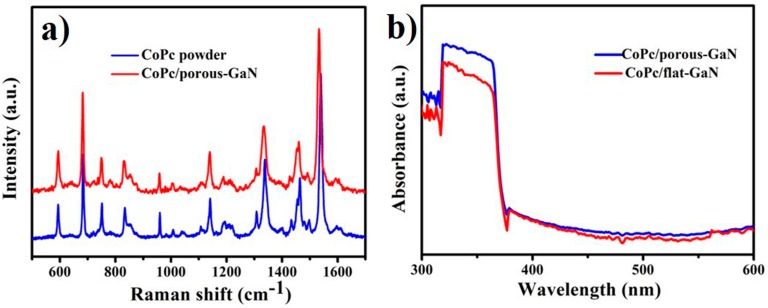
(**a**) Raman spectra of CoPc powder and CoPc/porous-GaN film; (**b**) Optical absorption spectra of CoPc/flat-GaN and CoPc/porous-GaN.

**Figure 4 nanomaterials-09-01198-f004:**
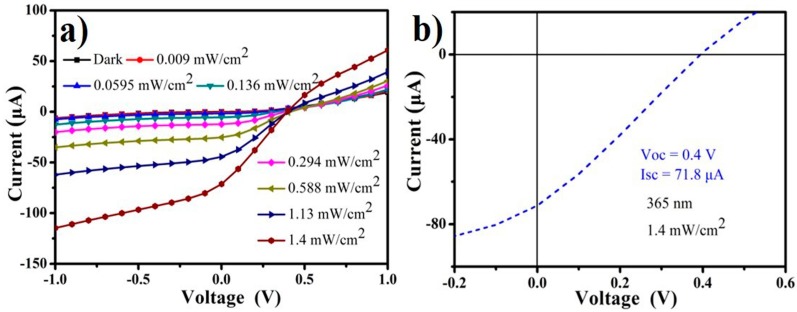
(**a**) Current–voltage (I–V) curves of the photodetector (PD) under 365 nm light with different power densities from 0 to 1.4 mWcm^−2^ illumination. (**b**) Photovoltaic effect of the corresponding device.

**Figure 5 nanomaterials-09-01198-f005:**
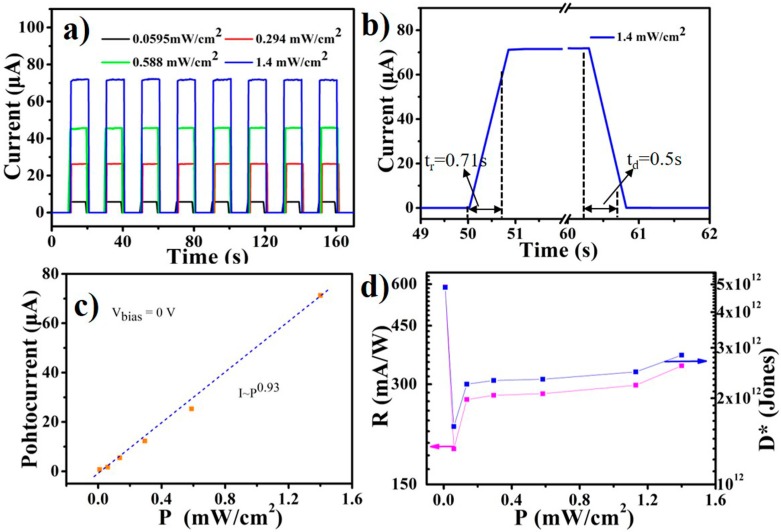
(**a**) Time-dependent on/off switching of the device with different power density light illumination. (**b**) The response time of the device under 1.4 mWcm^−2^ light illumination. (**c**) Power-density-dependent photocurrent and its corresponding fitting curve. (**d**) Power-density-dependent responsivity and specific detectivity. All the tests were carried out at 0 V bias and used 365 nm light illumination.

**Figure 6 nanomaterials-09-01198-f006:**
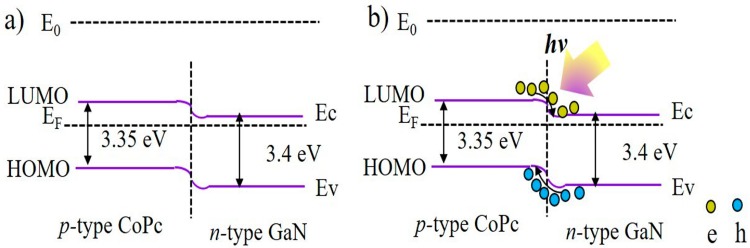
Band gap diagram of CoPc/porous-GaN *p*-*n* heterojunction UV PD under (**a**) the dark condition and (**b**) illumination condition.

**Figure 7 nanomaterials-09-01198-f007:**
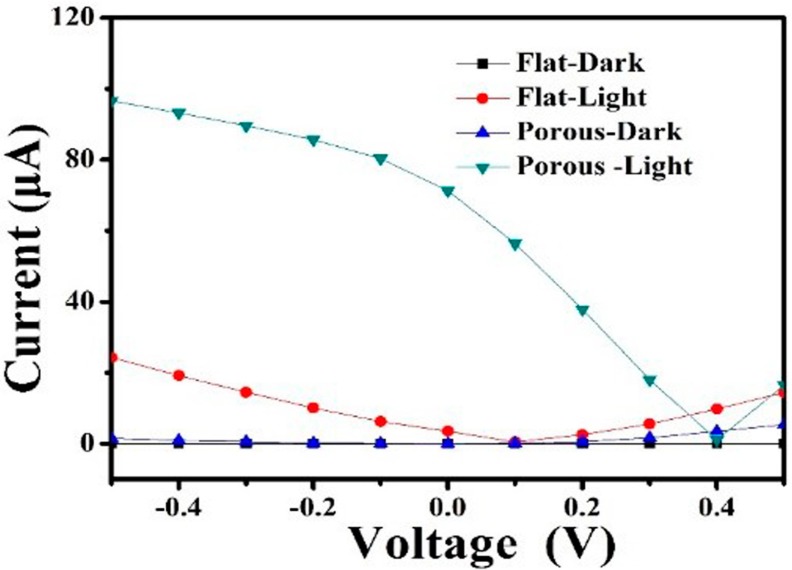
I–V curves of CoPc/porous-GaN- and CoPc/flat-GaN-based UV PDs.

**Table 1 nanomaterials-09-01198-t001:** Performance comparison between different self-powered GaN-based UV PDs.

Materials	Light Source	Switch Ratio	Responsivity (mA/W)	Detectivity (Jones)	Reference
GaN/Si- nanoporous pillar array	305 nm	~10^4^	29.4	-	[10]
MoS_2_/GaN	265 nm	~10^5^	187	2.34 × 10^13^	[11]
Ga_2_O_3_/GaN	365 nm	152	54.49	1.23 × 10^11^	[53]
n-GZO NRs/porous-GaN	365 nm	~10^5^	230	2.32 × 10^12^	[54]
CH_3_NH_3_PbI_3_/GaN	365 nm	5000	198	7.96 × 10^12^	[38]
CoPc/porous-GaN	365 nm	~10^5^	588	4.8 × 10^12^	This work

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
