# Peer review of "High-Performance Self-Powered Ultraviolet Photodetector Based on Nano-Porous GaN and CoPc p–n Vertical Heterojunction"

_nanomaterials, 2019, doi:10.3390/nano9091198_

Round 1

Reviewer 1 Report

Manuscript: 567976

Title: High-performance self-powered ultraviolet photodetector based on nano-porous GaN and CoPc p-n vertical heterojunction

By Yan Xiao et al.

In this paper the authors have realised and characterized a novel UV photodetector with self-powered ability on the basis of CoPc/porous-GaN p-n heterojunction. The paper is of interest and gives new idea for applications in the field of UV photodetectors.

However the paper can be considered for publication only after major revisions according to the critical points listed below.

Fig.1. The scheme in figure 1 does not contain relevant information on the device fabrication. The figure could be much improved with a cross-section view during the various steps of the fabrication process showing the real sizes of the device.

Fig.2. The labels a,c are not clearly visible.

In Fig. 2b concerning the percentage of the porous sizes it is not clear the reason for having 0% below 20nm and above 110nm. Moreover from the histogram the values of the percentage seem to be integer numbers. The authors should clarify in the text the statistical parameters used for this calculation.

Figure 2c: the cross section in the inset in figure 2c has a poor resolution. A better resolution is needed.

Line 147 and Line 209. Wrong numbering of sections: 3.3. Device Characterization, and 3.3. Sensing Mechanism. Corrections will have repercussions also on the section 3.4 Performance Enhance Mechansim

Line 190. “respoose” is misprinted

Eqs.(1) and (2). A reference for these simple equations is needed. The quantity R (responsivity) is not defined, The quantity “e” is not defined. Line 196. The quantity “Popt“ is defined as power density. But power density was “P” along the whole manuscript. Eq.(3) is misprinted. Probably the right equation is LDR= 20 log(Ip/Id)

Moreover the bibliography is rather incomplete and many important references should be added in the text showing the capability to enhance the UV-Blue absorption in GaN based nanostructures, opals etc... for example we suggest at least these listed below

G. Leahu et al., Study of thermal and optical properties of SiO2/GaN opals by photothermal deflection technique, Opt. Quant Electron 39: p. 305-310 (2007). Leahu, G. et al, Scientific Reports, 7, Article number 2833 (2017). Petronijevic, E, et al, International Journal of Thermophysics, 39, 45 (2018).

Author Response

First, we would like to thank the referees for their constructive comments which really helped in improving the manuscript. The comments and questions of the referees all have been taken into account in revising the manuscript as follows.

 1. The scheme in figure 1 does not contain relevant information on the device fabrication. The figure could be much improved with a cross-section view during the various steps of the fabrication process showing the real sizes of the device.

A new schematic of the fabrication process with cross-section view has been updated and the real size of the device was labeled.

2. The labels a,c are not clearly visible.

a and c of Fig.2 were clearly labeled in the manuscript that signed in the revised text.

3.In Fig. 2b concerning the percentage of the porous sizes it is not clear the reason for having 0% below 20nm and above 110nm. Moreover from the histogram the values of the percentage seem to be integer numbers. The authors should clarify in the text the statistical parameters used for this calculation.

In this work, we randomly selected 100 pores for size statistic analysis. The result given in Fig.2b show that among these 100 pores, there are no pores that are smaller than 20 nm or larger than 110 nm. Perhaps it is related with the etching condition used in this work. Also, to make the histogram of fig. 2b clearer, numbers for each bar was added in the figure. The corresponding change was added in the text as follows:

4. Figure 2c: the cross section in the inset in figure 2c has a poor resolution. A better resolution is needed.

Inset graph of figure 2c with higher resolution was replaced in the revised version.

5.Line 147 and Line 209. Wrong numbering of sections: 3.3. Device Characterization, and 3.3. Sensing Mechanism. Corrections will have repercussions also on the section 3.4 Performance Enhance Mechansim.

Thanks for your carefully checking. The wrong section number was corrected with red words in the revised version.

6.Line 190. “respoose” is misprinted

Line190 (line 201 in new version) "respoose" has corrected with "response" with red words.

7.(1) and (2). A reference for these simple equations is needed. The quantity R (responsivity) is not defined, The quantity “e” is not defined. Line 196. The quantity “Popt“ is defined as power density. But power density was “P” along the whole manuscript. Eq.(3) is misprinted. Probably the right equation is LDR= 20 log(Ip/Id)

The quantity R and e are defined in line of 207 and 208. The power density was unified defined as P in line 205 and 207. The misprinted Eq.(3) was corrected with LDR= 20 log(Ip/Id) in the revised text. All of the corrected places were signed with red words.

8.Moreover the bibliography is rather incomplete and many important references should be added in the text showing the capability to enhance the UV-Blue absorption in GaN based nanostructures, opals etc... for example we suggest at least these listed below G. Leahu et al., Study of thermal and optical properties of SiO2/GaN opals by photothermal deflection technique, Opt. Quant Electron 39: p. 305-310 (2007). Leahu, G. et al, Scientific Reports, 7, Article number 2833 (2017). Petronijevic, E, et al, International Journal of Thermophysics, 39, 45 (2018).

In order to make the manuscript more stringency, some important references were added with red words in the revise text.

“26. Leahu, G.; Voti, R. L.; Sibilia, C., Bertolotti, M.; Golubev, V.; Kurdyukov, D. A. Study of thermal and optical properties of SiO2/GaN opals by photothermal deflection technique. Opt. Quant. Electron. 2007, 39(4-6), 305-310.

Benedetti, A.; Alam, B.; Esposito, M.; Tasco, V., Leahu, G.; Belardini, A.Voti, R.L.; Passaseo, A.; Sibilia, C. Precise detection of circular dichroism in a cluster of nano-helices by photoacoustic measurements. Sci. Rep. 2017, 7(1), 5257. Petronijevic, E.; Leahu, G.; Belardini, A.; et.al. Photo-Acoustic Spectroscopy Reveals Extrinsic Optical Chirality in GaAs-Based Nanowires Partially Covered with Gold. International Journal of Thermophysics, 2018,39(4), 46.”

Reviewer 2 Report

The paper "High-performance self-powered ultraviolet photodetector based on nano-porous GaN and CoPc p-n vertical heterojunction" by Xiao et al. is a very interesting study on a novel self-powered UV PD developed by a p-n heterojunction via a thermal vapor deposition method. I strongly recommend its publication in Nanomaterials.

I would suggest a spell check of the text. Below is the list of some mistyping I detected:

line158 "deveice" instead of "device"

line 170 "exprosed" instead of "exposed"

line 176 "execellent" instead of "excellent"

line 190 "determinses" instead of "determines"

line 201 "aslo" instead of "also"

line 203 ":[11]" instead of "[11]:"

Author Response

Thanks for your recognition to this manuscript and carefully checking! All of the misprinting have corrected in the revised version with red sign.

line169 "deveice" has corrected with "device" line 181 "exprosed" has corrected with "exposed" line 187 "execellent" has corrected with "excellent" line 200 "determinses" has corrected with "determines" line 213 "aslo" has corrected with "also" line 215 ":[11]" has corrected with "[11]:"

Reviewer 3 Report

The manuscript presents the authors' work on the fabrication and characterization of a photodetector device comprising nano-porous GaN and CoPc p-n vertical heterojunction. The manuscript is well organized and their methods and analyses are clearly presented. The novelty of the device and the superiority of the device performance should be examined for the publication decision.

Clearly describe the novelty of the device in detail, with reviewing relevant past works.

I guess that the photodetector performance the authors obtained for their device may not be state-of-the-art values.

Reviewing the past reports on similar structured devices and also for all the category of GaN-based devices, clearly discuss how your device performance is positioned good or not.

The authors present a poor comparison variation, only with a CoPc/flat-GaN PD the authors prepared by themselves.

A good example for convictive performance comparisons is found in a Nanomaterials article:

https://www.mdpi.com/2079-4991/9/3/327

Author Response

1.Clearly describe the novelty of the device in detail, with reviewing relevant past works.

The novelty of our device clearly described as follows and added in the text with red words (from line 75 to line 85):

“In this work, we designed a novel high performance self-powered UV PD based on the organic/inorganic hybrids p-n heterojunction. Specific organic small conjugated molecules CoPc which have wide band gap (3.35 eV) and high thermal stability were used as p-type materials, and GaN with nano pores was employed as n-type material. A p-n heterojunction was formed at the interface of porous-GaN and CoPc to enhance the sensitivity and form a self-powered PD. The large specific surface area of nano porous structure not only improve light absorption, but also serve as photo traps to increase light absorption for its unique porous structure. The proposed device exhibited higher responsivity (R), more specific detectivity (D*), larger switch ratio (Ion/Ioff), and wider linear dynamic range (LDR) at 0 V bias to UV light compare to other single GaN based or inorganic/inorganic hybrids based UV PD. Furthermore, owning to the high thermal stability of GaN and CoPc, the photoelectrical property and on/off switching behavior of our device remained the same after 3 months.”

2.I guess that the photodetector performance the authors obtained for their device may not be state-of-the-art values.

Thanks for your challenging question. Although the detectivity of our device is lower than the MoS2/GaN inorganic hybrids PD, the responsivity and switch ratio are higher than most reported GaN based PD as compared in Table 1. In addition, considering the ultra-stability and low cost of CoPc, our PD may still have lots of potential values in real applications.

3.Reviewing the past reports on similar structured devices and also for all the category of GaN-based devices, clearly discuss how your device performance is positioned good or not.

A short discussion of the device performance position was added in the revised version. As follows:

“To compare with reported works, some features of self-powered GaN based UV PD were summarized in Table 1. From this table, the responsivity of our device is the highest among the listed works. The detectivity is about five-fold lower than Zhou’s work, but demonstrated similar or better level than other works. Also, for the switch ratio, our device can reach the best order of magnitudes among the table. Therefore, our device exhibited good performance in terms of responsivity, detectivity and switch ratio. ”

4. The authors present a poor comparison variation, only with a CoPc/flat-GaN PD the authors prepared by themselves. A good example for convictive performance comparisons is found in a Nanomaterials article: https://www.mdpi.com/2079-4991/9/3/327

Another important reference was added in the revised text. As follows:

“4.  Zhang, X.; Lin, Z.; Peng, D.; Diao, D. Bias-Modulated High Photoelectric Response of Graphene-Nanocrystallite Embedded Carbon Film Coated on n-Silicon. Nanomaterials, 2019, 9(3), 327.”

Round 2

Reviewer 1 Report

The manuscript has been revised according to the referee suggestions. It is now improved and ready to be published after 2 small corrections.

Figure 2b. The axis title "Pore size (nm)" is not clearly visible. Please correct also the value of the histogram bar "24" with "22" which allows to reach 100% Line 208. Please add the unity for the elementary charge (1.60 x 10-19 C)

Author Response

1. Figure 2b. The axis title "Pore size (nm)" is not clearly visible. Please correct also the value of the histogram bar "24" with "22" which allows to reach 100%.

A clearly Figure 2b has corrected and the value of the histogram bar "24" has corrected with "22". 

2.Line 208. Please add the unity for the elementary charge (1.60 x 10-19 C)

The unity of elementary charge (1.60 x 10-19 C) has added in line 208 with red words.

Reviewer 3 Report

The authors have addressed my concerns. 

For the point #4, I never meant that the article Zhang et al. should be cited. Zhang's paper is irrelevant to the authors' present manuscript, so remove its citation. I  meant that the authors should adopt the style Zhang et al. showed in their comprehensive performance-comparison tables.

Author Response

1. For the point #4, I never meant that the article Zhang et al. should be cited. Zhang's paper is irrelevant to the authors' present manuscript, so remove its citation. I meant that the authors should adopt the style Zhang et al. showed in their comprehensive performance-comparison tables.

I am sorry for the misunderstanding. Ref. 4 of Zhang's paper has removed in the revised text.
